# Sequence-to-sequence modeling for Temporal Reconstruction of Cellular Events

## Abstract

Single-cell omics technologies capture molecular snapshots of cells, while most biological processes unfold over time. Accurately predicting single-cell gene expression at unmeasured time points enhances our understanding of these processes, reducing costs and experimental effort by enabling the interpolation and extrapolation of observed data. This helps study continuous development, response to perturbations, and disease progression. To address this problem, we propose an encoder-decoder transformer architecture for Temporal Reconstruction of Cellular Events (TRACE). TRACE models gene expression generation as a sequence-to-sequence generation task by learning to transform a sequence of genes from a source condition (e.g., previous time) into a sequence of genes in a target condition (e.g., next time point). TRACE decoder learns to generate gene tokens of the target condition by iteratively unmasking tokens in the target sequence, overcoming the discordance between autoregressive modeling and the non-sequential nature of gene expression data. We evaluate TRACE both quantitatively and qualitatively on three datasets, covering a range of tasks and biological scenarios. TRACE outperforms existing models in generalizing across in-distribution and out-of-distribution tasks for temporal prediction. Furthermore, we demonstrate the biological relevance of the cell embeddings learned by TRACE by delineating activation-dependent cell stages in immune cells, measured across multiple time points. Our findings suggest that TRACE can enhance in silico hypothesis generation, improving our understanding and prediction of cellular changes over time. This ultimately facilitates disease understanding and supports the design of cost-effective experiments for biological discovery.

## 1 Introduction

Investigating how cells and tissues respond to external perturbations (i.e., interventions) such as drugs, biochemical stimuli, or gene editing is central to understanding (patho-)physiology and developing efficient therapeutics. In this context, single-cell RNA sequencing (scRNA-seq) provides a pivotal tool for transcriptomic profiling of cells at unparalleled resolution and scale (Svensson et al., 2020). However, scRNA-seq experiments are expensive and complex (Huang et al., 2024). Additionally, the destructive nature of the technology prevents repeated sampling from the same cell, which poses a challenge for studying continuous biological processes. Thus far, time-resolved single-cell studies are limited in the number of sampled time points and throughput due to the associated cost and logistical overhead (i.e. performing 24h time course experiments, limited availability of clinical samples). Generative machine learning methods have emerged as a promising avenue for inferring perturbation responses across time. Such *in silico* temporal predictions can support experimental design, scientific discovery and ultimately drug development.

Multiple computational frameworks have been developed to predict single-cell condition-specific gene expression. For example, generative modeling using variational auto-encoders (VAEs)(Kingma & Welling, 2014) combined with vector arithmetics (Lotfollahi et al., 2019) or disentanglement learning (Hetzel et al., 2022; Lopez et al., 2023). Optimal transport(Bunne et al., 2023; Huguet et al., 2022; Tong et al., 2024; Schiebinger et al., 2019; Klein et al., 2023a) and dynamical modeling (Tong et al., 2024; Huguet et al., 2022; Yeo et al., 2021) based methods have also yielded promising results. More broadly, these models predict single cell gene expression counts for missing conditions in time-

series prediction settings, or in response to perturbations such as drugs, diseases, and endogenous physiological stimuli (e.g., cytokines).

In parallel, large-scale masked language modeling (Devlin et al., 2018b; Achiam et al., 2023; Raffel et al., 2020a) has been applied to train single-cell foundation models (Cui et al., 2024; Theodoris et al., 2023). By analogy to natural language processing, cells (sentences) are treated as sequences of genes (words). In terms of perturbation response prediction, Geneformer examines the impact of removing genes from the cell sequence (analogous to an experimental knock-out) on cell embeddings (Theodoris et al., 2023; Chen et al., 2024) while scGPT has been specifically fine-tuned to predict unseen multi-gene perturbations (Cui et al., 2024).

In this work, we propose TRACE, the first sequence-to-sequence (seq2seq) encoder-decoder single-cell generative model designed to predict temporal changes in cells (Fig. 1), inspired by advances in seq2seq modeling in language and multi-modal learning (Raffel et al., 2020b; Yu et al., 2022; Chang et al., 2022). TRACE addresses the challenging task of predicting temporal changes in single-cell data. The model takes a sequence of gene tokens from a source condition (e.g., time point $t$) as input and generates a transformed sequence for a target condition (e.g., time point $t'$). This differs from existing methods (Bunne et al., 2023; Huguet et al., 2022; Tong et al., 2024; Schiebinger et al., 2019; Klein et al., 2023a), which rely on low-dimensional cell embeddings (e.g., PCA of the data) to directly generate gene-level embeddings for unseen time points. TRACE, on the other hand, generates gene-level embeddings, which allow for gene space analysis or easy conversion back to the original count space. More importantly, operating at the gene level enables the model to directly learn gene-gene relationships across time points.

TRACE functions as both a generative and an embedding model, unlike current encoder-only single-cell transformer models Cui et al. (2024); Theodoris et al. (2023). Its flexibility allows for the modeling of high-dimensional single-cell data without relying on dimensionality reduction, unlike recent innovations using flow matching and optimal transport, which primarily operate in low-dimensional spaces (Huguet et al., 2022; Yeo et al., 2021). TRACE's learned embedding space enables seamless transformation from token space to gene expression count space through a count decoder. Additionally, TRACE can be easily integrated into the foundation model pre-training stack and scales to large-scale pre-training, leveraging transformers' efficient and parallelizable training strategies developed in the NLP and LLM communities (Dao et al., 2022), while avoiding challenges in VAE training, such as posterior collapse (Dai et al., 2020) and provides an alternative to the promising flow matching and diffusion models in this space.

We demonstrate TRACE's abilities to predict condition-specific changes and support downstream analyses across comprehensive experiments. TRACE outperforms existing methods in both in-distribution and out-of-distribution prediction tasks for time-specific changes. TRACE effectively captures biological signals in cell embeddings, such as cell types and populations, and achieves superior performance for modeling count distributions. Finally, we highlight another use case showing how TRACE can capture known gene markers for T cell activation through gene embedding analysis, underpinning its potential to uncover novel biological processes.

## 2 RELATED WORKS

**Modeling cells as a sequence** The first model to represent cells as a sequence of tokens (genes) was scBERT(Yang et al., 2022), which used bidirectional encoder pretraining by masking gene labels, similar to BERT(Devlin et al., 2018a). Geneformer(Theodoris et al., 2023) introduced rank value encoding, where each genes' expression is normalized based on the median expression across a corpus of 30M cells, and then ranked within each cell. scGPT(Cui et al., 2024) uses autoregressive masking to predict gene expression binned values. However, no existing work represents the single-cell generation problem as a full seq2seq task using a transformer encoder-decoder formulation. The power of encoder-decoder architectures for generative modeling has been demonstrated in text generation(Raffel et al., 2020b), text-to-image(Yu et al., 2022), and audio(Borsos et al., 2023) models, motivating our current work.

**Modeling temporal dynamics** We are interested in the task of predicting gene expression at time point $t'$ given single cell gene expression at time point $t$ (and a potential perturbation). The temporal coupling between cell populations at time points t and $t'$ has been leveraged by multiple methods

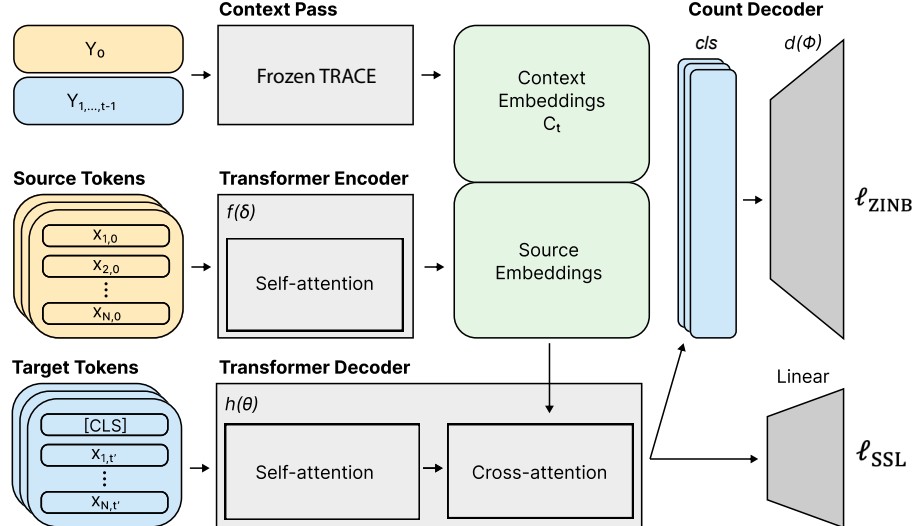

Figure 1: **TRACE architecture**. **a** The target sequence at time point $t'$ consisting of a gene tokens $x_{i,t'}$ and a [CLS] token for cell, where the remaining time points are provided as the context in the cross-attention. The trained context embeddings $C_t$ are retrieved using a forward pass through the transformer model. **b** Source and target cells are passed into an encoder $f(\delta)$ and decoder network $h(\theta)$, respectively. Here, each colored box represents a single tokenized cell. The model is optimized in a self-supervised manner by predicting the proportion of masked target tokens. **c**, Cell embeddings $cls$ are used to reconstruct gene counts of the target condition using a count decoder $d(\phi)$ optimized with a count loss $\ell_{ZINB}$.

treating this as an optimal transport (OT) problem. For example, developmental processes can be approximated by OT couplings, with transitions between progenitor and differentiated cell states modeled as locally linear transitions between probability distributions (Schiebinger et al., 2019). CellOT combines OT and input convex neural networks to learn OT maps in a fully paramterized manner, yielding improvements in scalability, stability and performance (Bunne et al., 2023). To account for non-linear trajectories in biological systems, TrajectoryNet incorporates continuous normalizing flows and a dynamic OT system (Tong et al., 2020). Conditional flow matching (CFM) generalizes this approach to arbitrary transport maps, avoiding the limitations of continuous normalizing flows (such as the assumption of deterministic process and a Gaussian starting distribution), and uses minibatch approximations to efficiently estimate the OT map in a simulation-free manner (OT-CFM) (Tong et al., 2023). MIOFlow uses neural ordinary differential equations (ODE) solver to learn an OT plan in a latent space which preserves geodesic distances between time points (Huguet et al., 2022). While further improvements to OT-based methods have been reported(Eyring et al., 2023), their applicability to the high-dimensional gene expression single-cell prediction tasks benchmarked here has not been demonstrated. Alternatively, PRESCIENT is a generative model which uses stochastic differential equations to model cellular differentiation as a diffusion process.

Here, we adopt a novel approach, modeling the time series task as a seq2seq problem. The advantage of this approach is that the model simultaneously learns the gene-level transformation of cells between time points and learns biologically-meaningful gene and cell embeddings. Additionally, unlike flow-based models, the seq2seq approach learns the transformation between the input distribution and target distribution (i.e., between the initial and final states) without requiring the explicit definition of closed-form conditional flows.

## 3 METHOD

TRACE is an encoder-decoder transformer designed to generate a sequence of genes and their expression for a cell under a desired target condition, given a sequence of genes in the source

condition. Uniquely, it combines context generation for the target condition with bidirectional masking. In temporal prediction, the representations of the other time points provide context for the generation. This approach enables the model to learn gene-gene relationships within a cell and across different conditions. In the following sections, we describe each component of our model in detail.

### 3.1 TRACE TRANSFORMER PRETRAINING

**Problem Formulation** Let $X_{t,j} = \{x_{i,t,j}\}_{i=1}^N$ denote gene tokens for a cell $j \in \{1, \ldots, L\}$ at time point $t$ where $N$ is the maximum number of tokens for each cell, $L$ is the number of cells and $t \in \{0, \ldots, T\}$. For simplicity, we omit the subscription of $j$ in the following formulations. We assign an embedding $Y_t = \{y_{i,t}\}_{i=1}^N$ where $y_{i,t} \in \mathbb{R}^d$ to each gene token. To learn a cell embedding, we introduce a unique special token [CLS] and prepend it to the sequence of gene tokens. We aim to generate cell and gene embeddings for a target time point $t'$ given all the remaining time points as context.

**Masking Strategy** During training, we randomly select a time point $t'$. Then, we sample a subset of $M$ tokens with a probability of $\beta$ from $X_{t'}$ based on masking scheduler function $\gamma$ and replace them with a [MASK] token following the MaskGIT (Chang et al., 2022) masking strategy. Since we have different sequence length padding, we need to ensure we do not mask $cl$ and pad tokens during training. So, we use an implementation trick (details in A.2 to prevent the masking of pad tokens). In the end, we get masked tokens $X'_{t'} = \{x_{i,t}\}_{k=1}^{M'}$ where $X'_{t'} \subset X_{t'}$ and $M'$ is the number of masked tokens.

**Training Objective** We feed the token embedding for the source time point $t_0$ to the transformer encoder $f$ with parameters $\delta$. The encoder generates the embedding $Z_0 = \{z_{i,0}\}_{k=1}^N$. Then, we pass the embeddings for the remaining time points to the transformer decoder $h$ with parameters $\theta$. The decoder is trained for time point $t'$, and the remaining time points and source are concatenated to generate the context embedding $C_t$. This provides context for the decoder's cross-attention. The training objective is to minimize the cross entropy loss for masked tokens:

$$\ell_{\text{pretraining}} = \sum_{t' \in [1, \ldots, T]} \sum_{i=1}^{M'} \log P(x_{i,t'} | \hat{X}_{\bar{M}, t'}, C_t) \tag{1}$$

where $\hat{X}_{\bar{M}, t'}$ are the remaining tokens after masking. This loss motivates the model to learn the cell and gene representation based on bidirectional masking. Attending to genes in both directions and different time points helps generate better cell and gene representations.

**Generating Context** For each time point $t$, excluding $t = 0$ and $t'$, we run the decoder in a forward pass without backpropagation to generate context embeddings. Context embeddings are used in the target sequence generation process described later. This process is autoregressive, meaning that the context embeddings for each $t$ are generated sequentially, using the embeddings from all previous time steps as context. The process starts by generating the embedding for the initial time step after the source time step ($t = 0$); subsequently, for each following time step, the newly generated embedding from the previous step is used as context. This continues iteratively until the context embedding for the last time step is generated. The context embedding at any time $t$ is given by the following equation:

$$C_t = h(Y_t \mid Z_0, \ldots, C_{t-1}) \tag{2}$$

**Gene expression decoder** Given the learned CLS embedding $cl_j$, the perturbed gene expression counts $G = \{g_{i,j}\}_{i=1}^R$ were predicted through a count decoder $d$ with parameters $\phi$, where $R$ is the number of genes. In detail, the count decoder is composed of a 2-layer multi-perceptron followed by Euclidean normalization and a zero-inflated negative binomial (ZINB) reconstruction loss, previously introduced by (Lopez et al., 2018). ZINB accounts for read dropout, an artifact of scRNA-seq data. (See Appendix 11 for more details.)

## 3.2 Cell sequence Generation

As in text generation, autoregressive decoding predicts tokens conditioned on the previously generated sequence. However, gene expression does not follow this unidirectional logic, as genes act together in non-sequential gene regulatory networks. Instead, the iterative decoder proposed in the bidirectional MaskGIT (Chang et al., 2022) transformer is more suitable to infer "cell sentences". While theoretically, this method could generate all tokens simultaneously, tokens are iteratively inferred as this approach yields superior results (Chang et al., 2022). The sequence starts blank with all unpadded tokens masked $\hat{x}_M^{(0)}$. At each iteration step $r$, a mask scheduling function $\gamma$ determines the number of masked tokens $n = \gamma\left(\frac{r}{R}N\right)$. As the number of iteration steps $r$ increases, the number of mask tokens decreases. The probabilities $p^{(r)} \in \mathbb{R}^N$ for the masked tokens $\hat{x}_M^{(r)}$ are predicted based on the bidirectional context of unmasked tokens. For each masked position, a token $x_i^{(r)}$ is sampled based on predictive probabilities $p_i$, wherein temperature annealing can be adjusted to modulate diversity. Moreover, gene tokens cannot occur multiple times within the same sequence, thus unmasked tokens are excluded from the possibilities. The remaining tokens undergo the same prediction cycle until the total step $R$ is reached, and all tokens are predicted.

**Interpolation and Extrapolation** We introduce two positional encodings. The first one captures the rank of gene tokens in a cell, and the second one determines the order of time points. We add the positional encodings $\boldsymbol{PE_{1,i}} = \{pe_{i,t}\}_{i=1}^N$ and $\boldsymbol{PE_{2,t}} = \{pe_{i,t}\}_{t=1}^T$ based on the position of each token within the cell's sequence, and the timepoint, respectively. We interpolate between two time points $t_{i-1}$ and $t_{i+1}$ by introducing new time points $t_i$ to the time positional encoding $PE_{2,t}$ between $PE_{2,t-1}$ and $PE_{2,t+1}$ during training. During testing, we also provide all time points as the context during generation to generate the interpolated time points. For extrapolation, we follow the training mode described above and provide all time points as the context during the generation. We can decide the sequence length and the number of time points by adjusting the positional encoding for both extrapolation and interpolation cells. We investigate the effect of using different types of positional encoding on the model's performance. (See the details and results at ablation 5.3.)

## 4 Experiments

### 4.1 Experimental setup

**Metrics** We use Maximum Mean Discrepancy (MMD, Mean kernel (Gretton et al., 2006)), Earth Moving Distance (1-Wasserstein, EMD (Cuturi, 2013)), Pearson correlation (PearsonR), and Rouge score(See et al., 2017). EMD is calculated for each gene separately based on (Lotfollahi et al., 2023), and the mean value over genes is reported. All metrics except the Rouge score are reported on log-normalized counts.

**Implementation** For the main model, we use a 6-layer transformer encoder-decoder. For the count decoder, we use a multi-layer perceptron with a GELU activation layer. We use Adam Optimizer (Kingma & Ba, 2015). We use NVIDIA A100 80 GB and H100 80 GB for all the experiments. See Appendix A.1 for more information about hyperparameters. To recreate other methods' results, we follow their respective repositories. We compute PCA on log-normalized counts to reduce the dimension to 100 dimensions for OT-CFM and 50 dimensions for other methods. For OT-CFM, we scaled the PC values as recommended. The predicted PC values are inverse-transformed to project back to the count space. You can find the repository here: https://anonymous.4open.science/status/TRACE-ICLR-3316

### 4.2 Datasets

**T cell** Soskic and Cano-Gamez, et. al. profiled single-cell gene expression of 655'349 naive and memory CD4$^+$ T cells from 119 donors which were measured at four time points (resting (0h) and $\alpha$-CD3,$\alpha$-CD28 activated T cells (16h, 40h and 5d)). Experimental procedures and scRNA-seq analysis steps (donor deconvolution, QC, cell type annotation) were kept the same as described in the publication (Soskic et al., 2022).

**Embryoid body** The embryoid body (EB) dataset is timely resolved to investigate the differentiation potential of human embryonic stem cells into distinct cell lineages. Over 27 days, samples were

acquired in 3-day time intervals for scRNA-seq. In total, 31'161 cells were analysed. After performing QC steps, 16'825 high-quality cells remained for downstream analysis (Moon et al., 2019).

**Lipopolysaccharide** The LPS (Lipopolysaccharide) dataset consists of CITE-seq data (Stoeckius et al., 2017) from 6 patients injected with LPS and their Peripheral Blood Mononuclear Cells (PBMCs) collected at 4 time points, 0min, 90min, 6hr (validation experiment) and 10hr. LPS is a component of bacteria when injected intravenously can elicit a controlled immune response similar to sepsis, a potentially serious condition resulting from a systemic and a dysregulated immune response to bacterial infection (van der Poll et al., 2017). In this study, the RNA modality of the data comprising 93'648 cells and 15 cell types are from volunteers injected with LPS and used as model to study sepsis. The data for time points 90min and 10hr has been published in (Stephenson et al., 2021).

### 4.3 PREPROCESSING

We use the ranked tokenization from Geneformer (Theodoris et al., 2023) to transform raw gene expression counts into a sequence of ranked gene tokens (Method 3.2). For the time-series experiments, gene features are filtered based on 2000 highly variable genes using Scanpy before tokenization (Wolf et al., 2019). A cell from source time point $t_0$ is paired to a cell from each target time point $t \in \{1, \ldots, T\}$ using either random or stratified pairing. Stratification conditions are used if reasonable experimental and biological anchors (e.g., perturbation, donor, cell types) exist. We add a cell pairing index to map gene tokens to the corresponding gene counts for count modeling. We use coarse cell type and donor in the case of T cell and only cell type for the LPS dataset as pairing condition.

### 4.4 RESULTS

Here, we show that TRACE obtains biologically meaningful cell and gene embeddings given ground truth gene tokens of source and different time points to study immune responses. Then the model's generative abilities are evaluated on time point interpolation and extrapolation. Lastly, we explore the dependency on the pretrained encoder and different components of the methods on the generation quality.

### 4.5 CELL AND GENE EMBEDDINGS RECOVER DISTINCT T CELL ACTIVATION STAGES

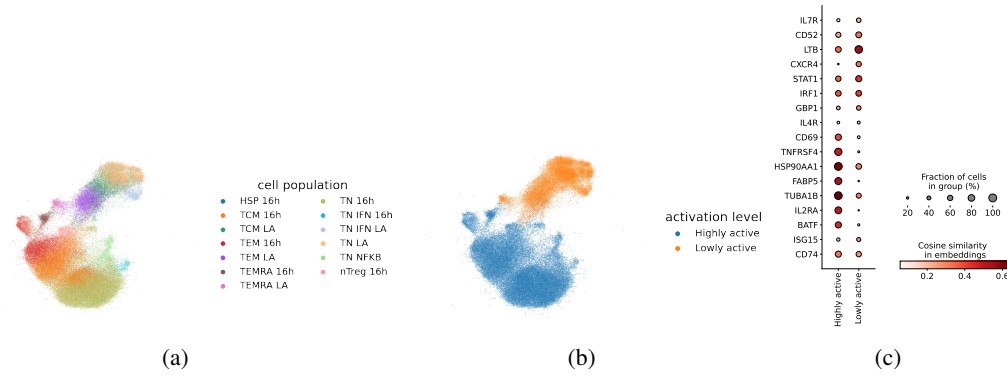

(a)                 (b)                 (c)

Figure 2: **Uniform Manifold Approximation and Projection (UMAP)** of cell embeddings of activated T cells at 16 hours colored by (a) granular cell types and (b) activation level. (c) Mean cosine similarity of cell embedding and gene embedding for activation level condition. The size of the dots indicates the proportion of cells expressing that gene.

In single-cell biology, cell and gene embeddings from deep learning models can be used to uncover cell states specific to biological conditions such as development and immune responses. T cell response to antigen stimulation is essential for triggering a healthy immune response. Characterizing this activation process can help detect genes involved in autoimmune diseases and cancer (Schmidt

et al., 2022; Soskic et al., 2019; 2022). We use TRACE to analyze the dynamics of T cell activation in a time-course of antigen-stimulated human CD4+ T cells. In this experiment, we train only the encoder-decoder transformer of TRACE in a self-supervised manner for 20 epochs and extract cell and gene embeddings. For each cell, we compute the cosine similarity between the cell and gene embeddings to determine which genes contribute most to the global cell representation. Even without supervised cell type information, in Figure 2 the cell embeddings recapitulate cell states defined by expert annotation in the original publication.

The authors report highly and lowly active T cell states with different transcriptomic profiles during early T cell activation. In accordance, the cell embeddings separate based on activation level. Additionally, the gene embeddings with highest cosine similarity to the highly activated T cells are known activation markers and cytokines such IL2RA, TNFRSF4 and CD69. Thus, both cell and gene embedding capture nuanced activation-dependent cell states in this highly homogeneous T cell population.

### 4.6 PREDICTING TEMPORAL IMMUNE RESPONSE TO BACTERIAL INFECTION

We qualitatively assess the generated gene expression for an interpolated time point at 6 hours (Figure 3) and an extrapolated time point at 10 hours after stimulation with LPS. TRACE distinguishes major immune cell types, including B cells, T cells, and monocytes, all of which have been previously reported to be affected at different stages (Ngkelo et al., 2012).. This shows that TRACE predicts the gene expression distributions across cell type during generation. (See Appendix B.4 for extrapolation).

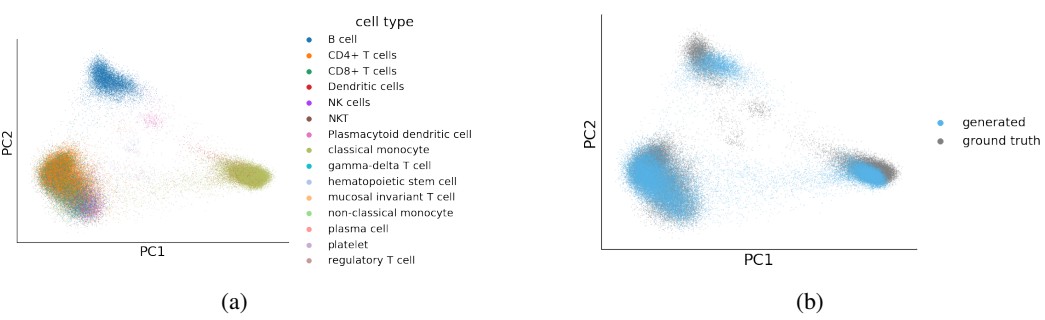

(a)                                                                              (b)

Figure 3: **Generated cells for the first time point for interpolation** (a) Cell type annotations of generated cells in the first two principal component spaces. (b) Generated cells for LPS treatment at 6 hours (LPS 6h) are overlaid onto true cells in the first two principal components (PC1 and PC2).

### 4.7 DISCRETE SINGLE CELL TEMPORAL INTER- AND EXRAPOLATION

In this experiment, we evaluate the generalization power of our method to unseen data in single-cell temporal interpolation and extrapolation. The EB dataset consists of time points $[0, 1, 2, 3, 4]$ while the T cell and LPS time points are $[0, 1, 2, 3]$. We exclude time point 3 for the EB dataset, time point 2 for the T cell dataset, and time point 1 from the LPS dataset, then generate cells for held-out time points for all datasets in the interpolation task. For extrapolation, we exclude time point 4 for the EB dataset, time point 3 for the T cell and the LPS datasets. The model trains on all the time points except those excluded for generation. (see Appendix B.2 and B.1 further details on gene marker and generated cell plots for T cell interpolation)

We compare our results with MIOFlow (Huguet et al., 2022), Prescient (Yeo et al., 2021) and OT-CFM (Tong et al., 2023). Table 1 and 2 show the results for interpolation and extrapolation; we use MMD and EMD for comparison. Prescient could not be applied to the T cell dataset because of poor scalability with the number of data points. TRACE outperforms all other methods for extrapolation and interpolation in three datasets based on EMD. OT-CFM performs similarly to TRACE in terms of MMD, which is known to be sensitive to differences in the mean values of distributions. In the case of scRNA-seq data, which is inherently sparse with a mean expression close to zero, OT-CFM

Table 1: **Held-out time point prediction for scRNA-seq time-series.** Interpolation performance was assessed based on MMD and EMD, and predicted and true expression values were compared. All results are reported over three random seeds.

| Method | EB (t=3) | | T cell (t=2) | | LPS (t=2) | |
|--------|----------|----------|--------------|----------|-----------|----------|
| | MMD ($\downarrow$) | EMD ($\downarrow$) | MMD ($\downarrow$) | EMD ($\downarrow$) | MMD ($\downarrow$) | EMD ($\downarrow$) |
| TRACE (ours) | **0.001±0.000** | **0.152±0.000** | **0.004±0.000** | **0.095 ± 0.006** | **0.001±0.000** | **0.152 ± 0.000** |
| MIOFlow | 0.061±0.004 | 0.207±0.000 | 0.082±0.008 | 0.119±0.009 | 0.034±0.002 | 0.269±0.005 |
| Prescient | 0.058±0.003 | 0.241±0.0001 | _ | _ | 0.032±0.000 | 0.488±0.003 |
| OT-CFM | **0.001±0.000** | 0.288±0.003 | **0.004±0.000** | 0.178±0.004 | 0.002±0.000 | 0.285±0.002 |

uses scaled PC (principle component) space, so their mean value is close to zero, so they perform similarly to TRACE in MMD, but TRACE outperforms OT-CFM significantly in EMD.

Table 2: **Held-out time point prediction for scRNA-seq time-series**. Extrapolation performance was assessed based on MMD and EMD, comparing predicted and true expression values. All results are reported over three random seeds.

| Method | EB (t=4) | | T cell (t=3) | | LPS (t=3) | |
|--------|----------|----------|--------------|----------|-----------|----------|
| | MMD ($\downarrow$) | EMD ($\downarrow$) | MMD ($\downarrow$) | EMD ($\downarrow$) | MMD ($\downarrow$) | EMD ($\downarrow$) |
| TRACE (ours) | **0.001±0.000** | **0.188±0.002** | 0.004±0.000 | **0.120 ± 0.006** | **0.001±0.000** | **0.188 ± 0.002** |
| MIOFlow | 0.062±0.007 | 0.212±0.005 | 0.106±0.006 | 0.16±0.006 | 0.033±0.003 | 0.288±0.008 |
| Prescient | 0.043±0.003 | 0.245±0.01 | _ | _ | 0.034±0.005 | 0.444±0.001 |
| OT-CFM | **0.001±0.000** | 0.380±0.002 | **0.002±0.000** | 0.287±0.006 | **0.001±0.000** | 0.726±0.021 |

# 5 ABLATION

## 5.1 TRANSFORMER ENCODER ANALYSIS

In Table 3, we evaluate three scenarios with the same training epochs. First, we evaluate the impact of using Geneformer as an encoder. In detail, we investigate frozen Geneformer and fine-tuned Geneformer compared to an encoder trained from scratch. Using a pretrained encoder is effective but not crucial since even training from scratch shows promising results. Furthermore, fine-tuning the pretrained encoder deteriorates the results; this could be due to the small size of the datasets.

Table 3: **Evaluation of Different Encoders**. Ablation study on different types of encoders based EMD$\downarrow$, MMD$\downarrow$ and PearsonR$\uparrow$.

| Encoder Type | MMD | EMD | PearsonR |
|--------------|-----|-----|----------|
| Pre-trained encoder (frozen) | 0.004 | 0.096 | 0.924 |
| Pre-trained encoder (fine-tuned) | 0.004 | 0.099 | 0.895 |
| Encoder from scratch | 0.008 | 0.102 | 0.836 |

## 5.2 IMPACT OF HYPERPARAMETERS ON GENERATION QUALITY

Figure 4 and Table 4 show the effect of hyperparameters on generation performance. Based on our experiments, the number of iterations only has a minor impact on the quality of generated cells, and the exponential scheduler shows the best performance during generation and training.

Furthermore, we investigate the effect of generated sequence length in Figure 5. Higher sequence length improves the Rouge score since the model has a higher chance of generating the correct genes. The identified optimal sequence length for the generation is in concordance with the actual mean sequence length 159. Thus, we use the mean length of the target sequences, excluding the prediction time points for the experiments. (See Appendix B.4 for more details on the effect of sequence length)

## 5.3 EXPERIMENTS FOR DIFFERENT TYPES OF POSITIONAL ENCODINGS

We investigate three different positional encoding scenarios to capture gene-gene relation in a cell and over time based on Method 3.2: **Sinusoidal Positional Encoding for Both Gene Rank and**

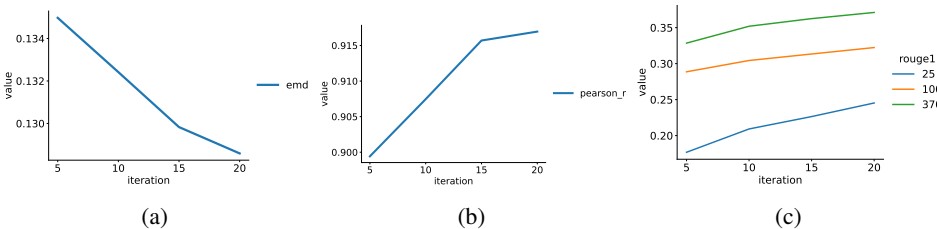

(a)          (b)          (c)

Figure 4: **Study of the number of iteration** (a, b) Number of iterations to generate samples based on EMD↓ and Pearson correlation↑, (c) rouge score↑ of three sequence lengths as the number of iterations increases.

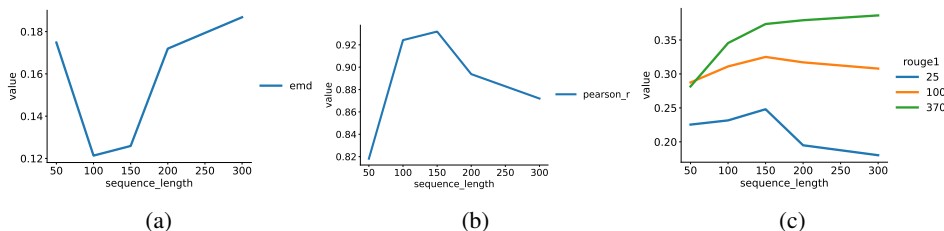

(a)          (b)          (c)

Figure 5: **Study of the sequence length** (a, b) Analysis of the effect of sequence length based on EMD↓ and Pearson correlation↑, (c) rouge score↑ of three sequence lengths for different sequence lengths as sequence length increases.

Table 4: **Evaluation of Different Schedulers**. Ablation study on different types of Scheduler method based on EMD↓, MMD↓ and PearsonR↑.

| Scheduler method | MMD | EMD | PearsonR |
|---|---|---|---|
| Cosine | 0.005 | 0.182 | 0.815 |
| Exponential | 0.005 | 0.156 | 0.841 |
| Cubic | 0.005 | 0.181 | 0.818 |

**Time:** In the first scenario, we use two different sinusoidal positional encodings for both, the gene rank within each cell and the time points. This approach leverages fixed positional patterns to capture structural information across genes and time points. **Learnable Positional Encoding for Gene Rank and Sinusoidal for Time:** In the second scenario, we use a learnable positional encoding for the gene rank within each cell, allowing the model to adaptively learn optimal positional representations for genes. Simultaneously, we use sinusoidal positional encoding for time positional encoding. **Unified Sinusoidal Positional Encoding Across Combined Time Points:** In the final scenario, we treat all time points as a single sequence. We use a unified sinusoidal positional encoding across this extended sequence, enabling the model to capture long-range temporal dependencies and interactions between genes over time. Table 5.3 shows performance for different scenarios, and the second approach shows the best performance.

## 6 CONCLUSION

**Discussion** In this paper, we introduce TRACE, a seq2seq transformer model designed for single-cell temporal prediction. We compare our approach with the state-of-the-art (SOTA) models in single-cell temporal data generation, and TRACE shows promising results across the tasks. We evaluate our model across three different studies in developmental biology, T cell activation, and response to infection. TRACE can generate embeddings of cells and genes at unseen time points, enabling analysis in embedding space while also demonstrating recoverability of original gene expression counts. These applications highlight the generative potential of TRACE. We envision that TRACE can facilitate temporal analysis of single-cell data and guide experimental design and cell engineering.

Table 5: **Evaluation of Different positional encoding**. Ablation study on different types of positional encoding based on EMD↓, MMD↓ and PearsonR↑. The first positional encoding is for time positional encoding, and the second is for gene positional encoding. The last option uses one sinusoidal over all time points length together

| Scheduler method | MMD | EMD | PearsonR |
|---|---|---|---|
| Sinusoidal+Learnable | 0.005 | 0.168 | 0.798 |
| Sinusoidal+Sinusoidal | 0.005 | 0.233 | 0.684 |
| Sinusoidal | 0.006 | 0.216 | 0.680 |

**Limitation** The effectiveness and performance of TRACE depend on how the data is paired. To capture cellular heterogeneity and cellular processes (i.e., temporal effects), having paired data is essential. However, the one-to-one mapping of a source cell in time point $t-1$ to a cell in time point $t$ does not capture biological processes such as cell growth and death which have been addressed in unbalanced OT (Schiebinger et al., 2019).

**Future work** Leveraging recent developments in NLP for fine-tuning (Zhao et al., 2024; Dettmers et al., 2024) and continual learning (Wang et al., 2024) of LLMs can improve the generative power of TRACE to enhance generation quality for unseen and rare cell types. Furthermore, this work can be applied to other areas of biology, such as developmental biology (Schiebinger et al., 2019; Klein et al., 2023b) or disease progression to study genes driving cellular changes or differentiation trajectories.

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

# A HYPERPARAMETERS AND IMPLEMENTATION DETAILS

## A.1 HYPERPARAMETERS

Hyperameters for interpolation for LPS dataset is in Table 6 and for extrapolation is in Table 7. For the T cell and EB dataset, we use the same hyperparameters for interpolation and extrapolation which is in Table 8 and in Table 9, respectively. For all the models, we load pre-trained Geneformer (gf-12L-95M-i4096 from (April 2024)) as the encoder transformer from HuggingFace and freeze all the layers during training and testing. This is referred to as 'Frozen Geneformer encoder'.

Table 6: **Hyperparameters for LPS Interpolation**

| Component | Parameter | Default Value |
|---|---|---|
| **General** | Batch size | 64 |
| **Transformer** | Learning rate | $1 \times 10^{-4}$ |
| | Weight decay | $1 \times 10^{-4}$ |
| | Masking probability | 0.15 |
| | Embedding size | 128 |
| | Frozen Geneformer encoder | Yes |
| | Number of attention heads | 8 |
| | Number of attention layers | 6 |
| | Attention head dimension | 64 |
| | Maximum sequence length | 647 |
| **Count Decoder** | Learning rate | $5 \times 10^{-3}$ |
| | Weight decay | $1 \times 10^{-3}$ |
| | Number of hidden layers | 2 |
| | Layer dimension | 128 |
| **Mask Decoder** | Temperature | 1.5 |
| | Iterations | 19 |
| | Mask scheduler | Cosine |

Table 7: **Hyperparameters for LPS Extrapolation**

| Component | Parameter | Default Value |
|---|---|---|
| **General** | Batch size | 64 |
| **Transformer** | Learning rate | $1 \times 10^{-4}$ |
| | Weight decay | $1 \times 10^{-4}$ |
| | Masking probability | 0.3 |
| | Embedding size | 32 |
| | Frozen Geneformer encoder | Yes |
| | Number of attention heads | 8 |
| | Number of attention layers | 6 |
| | Attention head dimension | 64 |
| | Maximum sequence length | 647 |
| **Count Decoder** | Learning rate | $5 \times 10^{-3}$ |
| | Weight decay | $1 \times 10^{-3}$ |
| | Number of hidden layers | 2 |
| | Layer dimension | 128 |
| **Mask Decoder** | Temperature | 1.5 |
| | Iterations | 19 |
| | Mask scheduler | Cosine |

Table 8: **Hyperparameters for T cell**

| Component | Parameter | Default Value |
|---|---|---|
| **General** | Batch size | 64 |
| **Transformer** | Learning rate | $1 \times 10^{-5}$ |
| | Weight decay | $1 \times 10^{-5}$ |
| | Embedding size | 512 |
| | Frozen Geneformer encoder | Yes |
| | Number of attention heads | 8 |
| | Number of attention layers | 6 |
| | Attention head dimension | 64 |
| | Maximum sequence length | 300 |
| **Count Decoder** | Learning rate | $5 \times 10^{-3}$ |
| | Weight decay | $1 \times 10^{-4}$ |
| | Number of hidden layers | 2 |
| | Layer dimension | 512 |
| **Mask Decoder** | Temperature | 0.5 |
| | Iterations | 20 |
| | Mask scheduler | Cosine |

Table 9: **Hyperparameters for EB**

| Component | Parameter | Default Value |
|---|---|---|
| **General** | Batch size | 64 |
| **Transformer** | Learning rate | $1 \times 10^{-3}$ |
| | Weight decay | $1 \times 10^{-4}$ |
| | Embedding size | 512 |
| | Frozen Geneformer encoder | Yes |
| | Number of attention heads | 8 |
| | Number of attention layers | 6 |
| | Attention head dimension | 32 |
| | Maximum sequence length | 270 |
| **Count Decoder** | Learning rate | $5 \times 10^{-4}$ |
| | Weight decay | $1 \times 10^{-4}$ |
| | Number of hidden layers | 2 |
| | Dropout | 0.25 |
| | Layer dimension | 512 |
| **Mask Decoder** | Temperature | 0.5 |
| | Iterations | 20 |
| | Mask scheduler | Cosine |

## A.2 IMPLEMENTATION DETAILS

**Implementation of Masking** We demonstrate the details of the masking strategy in the following Algorithm. The masking idea is adapted based on MaskGIT (Chang et al., 2022). We prevent padding tokens by adding line 7 to the algorithm so padding tokens get the highest probability; therefore, they don't get chosen for the masking.

---

**Algorithm 1:** Masking algorithm

---

**Input:** $pad$, $input\_id$, $mask\_scheduler$, $mask\_token$
**Output:** $input\_id$, $labels$

1  $sample\_length \leftarrow$ sum of non-padding tokens in $pad$ ;
2  $batch$, $seq\_len \leftarrow$ shape of $input\_id$;
3  $rand\_time \leftarrow$ uniform random values of size $(batch)$;
4  $rand\_mask\_probs \leftarrow$ noise schedule of $rand\_time$;
5  $num\_token\_masked \leftarrow$ round($sample\_length \times rand\_mask\_probs$);
6  $rand\_int \leftarrow$ random values of size $(batch,\ seq\_len)$;
7  Set padding positions in $rand\_int$ to 1;
8  $batch\_randperm \leftarrow$ argsort of $rand\_int$;
9  $mask \leftarrow batch\_randperm < num\_token\_masked$;
10 $input\_id[mask] \leftarrow mask\_token$;
11 Update labels: $labels[\neg mask] \leftarrow -100$;

---

**Implementation of ZINB loss** The Zero-Inflated Negative Binomial (ZINB) loss function is defined in the following. We used the implementation from SCVI (Lopez et al., 2018):

$$\ell(g; \mu, \theta, \pi) = -\mathbb{I}[g = 0] \cdot \ln\left(\pi + (1-\pi)\left(\frac{\theta}{\theta+\mu}\right)^{\theta}\right)$$

$$-\mathbb{I}[g > 0] \cdot \left(\ln(1-\pi) + \ln\binom{g+\theta-1}{g} + g\ln\left(\frac{\mu}{\theta+\mu}\right) + \theta\ln\left(\frac{\theta}{\theta+\mu}\right)\right) \tag{3}$$

Where $g$ is the observed count, $\mu$ is the mean of the Negative Binomial distribution, $\theta$ is the dispersion parameter (overdispersion), $\pi$ is the probability of zero inflation (dropout probability), and $\mathbb{I}[\cdot]$ is the indicator function, which equals 1 when the condition is true and 0 otherwise. $\binom{n}{k}$ is the binomial coefficient, defined as $\binom{n}{k} = \frac{\Gamma(n+1)}{\Gamma(k+1)\Gamma(n-k+1)}$.

### A.3 GENERATION DETAILS

## B EXPERIMENTAL RESULTS

### B.1 TCELL GENERATED CELL EMBEDDINGS FOR INTERPOLATION

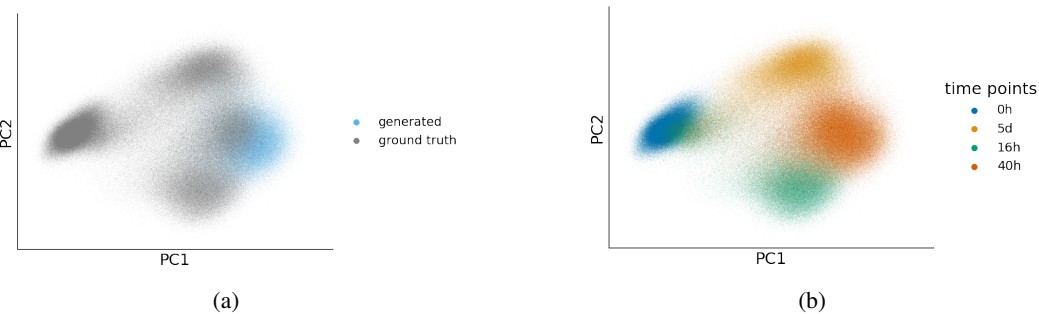

(a)
(b)

Figure 6: **Generated cells for 40h timepoint and ground truth for all timepoints for T cell dataset** (a) colored based on generated and ground truth for two principal components (PC1 and PC2), (b) colored based on different time points two principal components (PC1 and PC2).

## B.2   T cell generated cell gene marker analysis for Interpolation

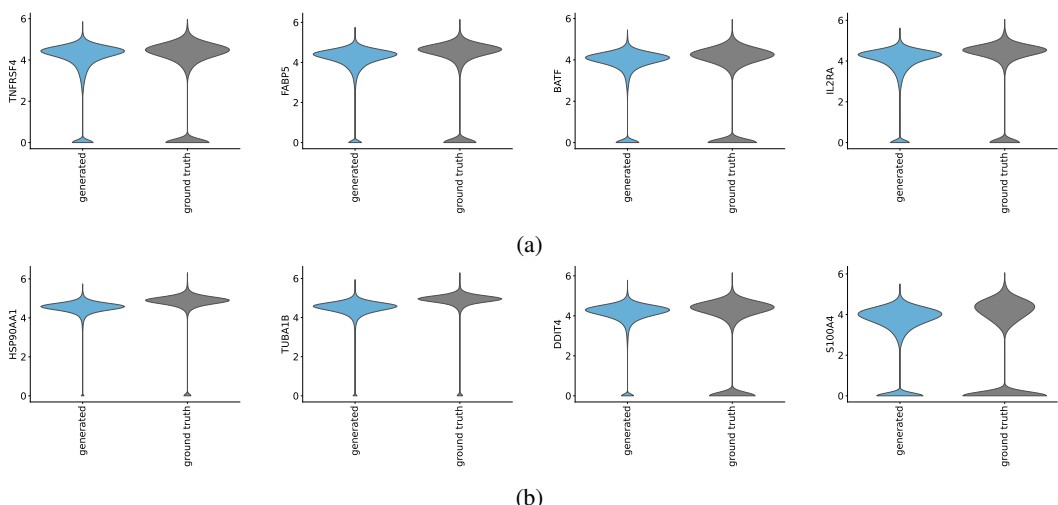

(a)

(b)

Figure 7: **Violin plot of activation gene markers at 40h time point comparing true to interpolated gene expression counts.** (a,b) Predicted log-normalized gene expression counts colored in blue contrasted with the ground truth counts. T cell activation-dependent gene markers (TNFRSF4, FABP5, BATF, IL2RA, HSP90AA1, TUBA1B, DDIT4 and S100A4) are shown, highlighted by the author in the original publication (Soskic et al., 2022)

## B.3   Tcell generated cells for imputation

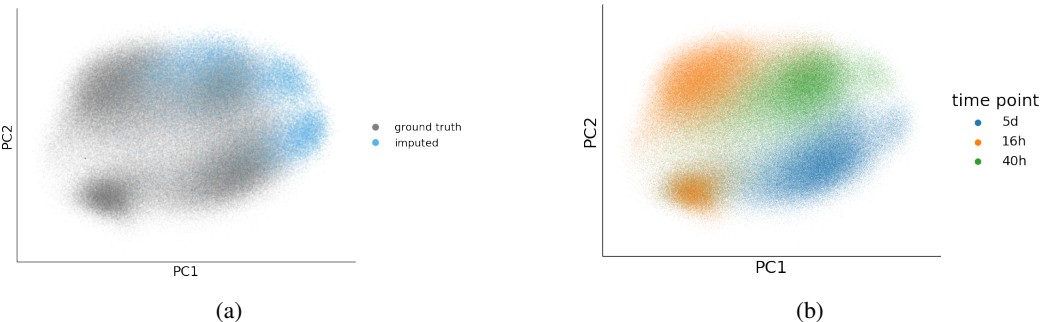

(a)                                                    (b)

Figure 8: **Imputed cells across all activated timepoints for in-distribution held-out cells (80%-20% train-test split) for T cell dataset** (a) colored based on generated and ground truth for two principal components (PC1 and PC2), (b) colored based on different time points two principal components (PC1 and PC2).

## B.4   LPS generated cell embeddings for extrapolation

9b shows the generated cell embeddings in PC space for the extrapolation of the LPS dataset. The generated cell captures the underlying distribution, but the information about some rare cell types is lost.

**Experiments for quality of extrapolation as the context length increases** We evaluate the capability of our method in extrapolation for the EB dataset. First, we train the model for the first two time points; then the trained model is used to extrapolate the three subsequent time points separately. As shown in Figure 10, we observe that as the time distance increases, the embedding quality decreases, leading to an increase in MMD. Pearson correlation does not change significantly as the time distance

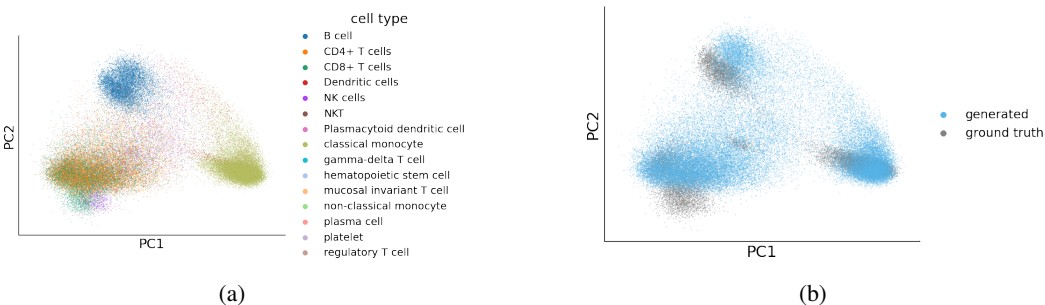

(a)                                          (b)

Figure 9: **Generated cells for the second timepoint for extrapolation** (a) Cell type annotations of generated cells in the first two principal component spaces. , (b)Generated cells for LPS treatment at 10 hours (LPS 10h) are overlaid onto true cells in the first two principal components (PC1 and PC2).

increases, likely due to the abundance of zero values in the raw counts. It makes it less sensitive to changes as long as zeros are predicted correctly.

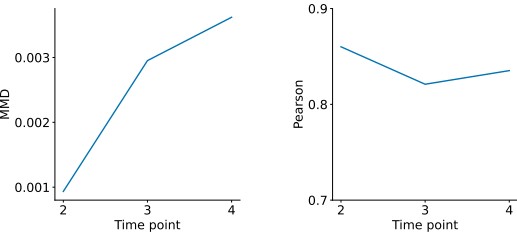

Figure 10: **Evaluation of extrapolation with increasing timesteps as distance to the source increases.** The left figure shows results for MMD, and the right figure shows results for Pearson correlation. The lower number is better for MMD, and a higher number is better for Pearson correlation.

