# OpenReview forum: "Sequence-to-sequence modeling for Temporal Reconstruction of Cellular Events"
_ICLR.cc/2025/Conference — ICLR 2025 Conference Withdrawn Submission_

### Official Review · Reviewer_sYbB · 2024-11-02

**Soundness:** 2
**Presentation:** 2
**Contribution:** 2
**Rating:** 3
**Confidence:** 2

**Summary:**

This paper introduces TRACE, an encoder-decoder transformer architecture aimed at temporal reconstruction of cellular events by modeling gene expression generation as a sequence-to-sequence task. In this setup, given a source single-cell gene expression sequence at a particular time point, the model generates a target sequence for a subsequent time point. Due to the non-autoregressive nature of single-cell data, TRACE employs an iterative unmasking generation style rather than the traditional autoregressive generation common in transformers. TRACE's efficacy is demonstrated across three biological datasets, highlighting its ability to generate data both in-distribution and out-of-distribution.

**Strengths:**

1. The authors apply sequence-to-sequence modeling techniques to modeling temporal dynamics of the single-cell data.
2. The proposed method doesn’t suffer from high dimensionality which certain methods like OT-CFM suffers.

**Weaknesses:**

Some sections need better clarity, especially the Method section. Here are a few areas where additional explanation might be helpful:

- The training and inference pipeline could be more thoroughly described, and Figure 1 could more closely align with the text description. A few specific questions came to mind:
  - How is the embedding \(Y_t\) assigned?
  - What is the process for generating gene tokens \(X_t\)?
  - In Figure 1, should the target tokens be masked?

- In addition, in the Introduction, the authors list several advantages of TRACE:
   1. It handles high-dimensional single-cell data without dimensionality reduction.
   2. It provides a learned embedding space for transformation between token and gene expression spaces.
   3. It integrates well with foundation model pretraining and offers scalability.
   4. It avoids challenges like posterior collapse often seen in VAE training.
  5. Its learned embedding space may benefit diffusion and flow-matching models.

However, only the first three points seem to be directly supported by experimental results. Providing further experimental validation for points (d) and (e) would strengthen the paper.

**Questions:**

- Could the authors expand on some of the methodological points mentioned in the “Areas for Improvement” section?
- The iterative unmasking approach in TRACE bears a resemblance to discrete diffusion with an absorbing transition matrix. Did the authors explore this connection?

---

### Official Review · Reviewer_5oUn · 2024-11-04

**Soundness:** 2
**Presentation:** 1
**Contribution:** 2
**Rating:** 3
**Confidence:** 3

**Summary:**

Understanding the evolution of cellular processes over time is crucial. Single-cell gene expression data collected at various time points provides an avenue to unravel these processes. However, collecting this data at many different time points is expensive. To help alleviate this problem, the authors develop a seq2seq model called TRACE. TRACE follows recent work such as Geneformer to encode a cell’s expression profile at any given time point as a sequence of gene tokens. Then, it uses an encoder-decoder architecture to predict the gene tokens at a target time point given the gene tokens at either previous time points (extrapolation) or surrounding time points (interpolation). The authors show that TRACE outperforms existing methods for predicting expression profiles at unseen time points.

**Strengths:**

- **Originality:** The main novel contribution of this work is the TRACE model for predicting single-cell gene expression profiles at unseen time points using data from other time points. Its main innovation is the usage of seq2seq modeling for directly capturing gene-gene interactions across time points instead of relying on cell embeddings as intermediates like in existing methods. Related work has been adequately cited and the authors state how TRACE improves upon existing methods.
- **Quality:** The authors benchmark TRACE against relevant baselines on three different datasets and show that it outperforms them for interpolation and extrapolation. However, their other analyses lack a compelling rationale in my opinion.
- **Clarity:** In the introduction and related work sections, the authors clearly articulate the motivation for interpolating and extrapolating temporal data from single-cell RNA sequencing (scRNA-seq). They also describe how seq2seq modeling can enhance temporal reconstruction. However, the remaining sections lack clarity and do not provide sufficient information to comprehend the specifics of the task and the methodology employed. In my opinion, the manuscript requires substantial revision for acceptance.
- **Significance:** TRACE outperforms existing methods for temporal reconstruction. However, due to issues with the clarity of the manuscript, it would be difficult for other researchers to understand and build upon this work. While the primary results comparing TRACE to baseline methods are compelling, the supporting results lack convincing evidence and do not align with the narrative.

**Weaknesses:**

Beginning with the Methods section, the clarity of this paper needs to be significantly improved for acceptance. Additionally, some of the analyses presented in the paper lack a clear motivation. Below are my suggestions for each section:

- **Methods:**
   - Although the general problem is stated in the introduction, a more detailed description is required. The current problem description lacks clarity and specificity. It fails to provide precise details about the inputs and expected outputs for interpolation and extrapolation, leaving the reader to make assumptions. It might be useful to illustrate the problem using one of the datasets used for evaluation.
   - The description of the training process is unclear and disjointed. The authors should provide a comprehensive, step-by-step guide outlining the training methodology. Providing an algorithm that describes the process would be useful.
   - It is also unclear how TRACE is utilized for inference. The concise descriptions in lines 237-241 are not particularly informative. Both interpolation and extrapolation appear to be performed in the same manner. Furthermore, it is unclear how the sequence length and number of time points can be adjusted, as they should be constants for a given dataset. Again, it would be beneficial if the authors could provide a succinct algorithm outlining the inference process for both interpolation and extrapolation.

- **Experiments:**
   - The metrics employed in this work should be meticulously described to ensure that individuals unfamiliar with the specific metrics utilized in prior work can comprehensively understand them.
   - In my opinion, the analysis presented in section 4.5 does not align with the overall narrative of the text. It fails to provide any insights into the temporal reconstruction capabilities of TRACE.

- **Ablation:** I find the analyses in this section very uninformative and it is unclear why these specific ablations were performed. It would be beneficial if authors could explain for why these ablations are probing the most critical aspects of their method. In my opinion, a more insightful ablation could be conducted on the training data used for interpolation/extrapolation. For instance, it would be interesting to explore how performance is affected by excluding data points that are closest to unseen time points.

**Questions:**

In addition to the suggestions above, I have the following questions:

- Why is $t_0$ data exclusively utilized for computing source embeddings? How is data from other time points incorporated into the training process?
- Why are only the source embeddings computed using the encoder? Why are the other embeddings computed using a frozen TRACE decoder?
- Do the encoder and decoder use the same gene token embeddings?
- It is unclear how the $Y_i$s are computed, are they computed using the decoder’s gene token embeddings?
- It also unclear what the $C_t$s represent, are they the gene embeddings for time $t$, the [CLS] token embeddings, or some combination/function of both?
- One of the main motivations for this work is to directly learn gene-gene interactions between time points to predict expression profiles at unseen time points. Although the seq2seq model accomplishes this objective to a certain extent, all final gene counts are still predicted using the cell embedding. Wouldn’t it be more effective to directly use the gene embeddings to make predictions instead of using the cell embedding?
- How are the two losses ($\ell_{SSL}$ and $\ell_{ZINB}$) combined during training?
- How exactly are the positional encodings for time computed? Lines 235-237 are not very clear about how they are computed.

---

### Official Review · Reviewer_4kyY · 2024-11-04

**Soundness:** 3
**Presentation:** 3
**Contribution:** 3
**Rating:** 5
**Confidence:** 4

**Summary:**

The authors present TRACE, a transformer-based model to infer temporal reconstruction of single cells. The authors analyzed the performance of the proposed model on three datasets and compared it to three previous models, showing (generally) superior performance. Additionally, the authors perform ablation studies on the different (hyper)parameters of TRACE, justifying the chosen architecture. TRACE is able to recapitulate known biology in the context of Immune response.

**Strengths:**

The authors proposed a very relevant model based on transformers to infer intermediate single cell states from scRNA-Seq data. The paper is well written and very clear. The architecture is well justified and several datasets and metrics are used to assess it. I appreciate the effort show that the model is able to recapitulate known biology in the context of immune response.

I also appreciate the use of principal components rather than the UMAP plots to qualitatively assess the overlap between the inferred and the ground truth data (Figure 3). The ablation studies are a nice addition to  better understand the proposed model.

**Weaknesses:**

While the proposed model is very relevant, I believe that the introduction is a bit misleading. The authors talk lengthy about in silico perturbation models (first three paragraphs ), but they only show results in data interpolation and ablation studies. It would be great if they could run experiments on scPerturb-Seq data such as those shown in the work: "Identifiability Guarantees for Causal Disentanglement from Soft Interventions" or the one cited in the paper by Lotfollahi et al. 2019. Inferring unseen (genetic) perturbations, as mentioned in the introduction, is a very relevant topic in Representation Learning applied to biomedicine/health. Thus, I believe that these experiments are of utmost importance to show the relevance of TRACE.

**Questions:**

How would TRACE work on scPerturb-Seq experiments?

---

### Official Review · Reviewer_Ln56 · 2024-11-05

**Soundness:** 2
**Presentation:** 1
**Contribution:** 2
**Rating:** 3
**Confidence:** 3

**Summary:**

This paper presents a novel architecture for generating temporal single cell trajectories. The authors train their model using several state-of-the-art techniques, including masked language modeling (i.e., mask out certain tokens from the gene and predicting), cross-attention, and zero-inflated negative binomial losses (zero-inflated to reflect the fact that gene expression counts are often zero-inflated (e.g., https://genomebiology.biomedcentral.com/articles/10.1186/s13059-015-0805-z). The authors test their framework on a number of single cell trajectory datasets. On these datasets, they perform comparably to or better than several competitor benchmarks.

**Strengths:**

The paper thoroughly documents their experiments and hyperparameters.

**Weaknesses:**

Comparison to baselines is poorly motivated and lacking. I don't understand why the measures used are used (why EMD, MMD, etc.? what are the distributions over?). Why don't you just use MSE of the full vector of gene expression between predicted state and true state? Why are only four time points reported in the paper (other time points don't seem to appear in appendix)?

**Questions:**

See above for baseline comparisons.

Q1) What actually is a "gene token" here? How are genes tokenized?

Q2) How does this approach compare to RNA velocity? Please add an RNA velocity baseline or explain why it's not possible.

---

### Note · Authors · 2024-11-27

I have read and agree with the venue's withdrawal policy on behalf of myself and my co-authors.